# Universal exploration dynamics of random walks

Léo Régnier[1], Maxim Dolgushev [1], S. Redner[2] & Olivier Bénichou [1]✉

The territory explored by a random walk is a key property that may be quantified by the number of distinct sites that the random walk visits up to a given time. We introduce a more fundamental quantity, the time $\tau_n$ required by a random walk to find a site that it never visited previously when the walk has already visited $n$ distinct sites, which encompasses the full dynamics about the visitation statistics. To study it, we develop a theoretical approach that relies on a mapping with a trapping problem, in which the spatial distribution of traps is continuously updated by the random walk itself. Despite the geometrical complexity of the territory explored by a random walk, the distribution of the $\tau_n$ can be accounted for by simple analytical expressions. Processes as varied as regular diffusion, anomalous diffusion, and diffusion in disordered media and fractals, fall into the same universality classes.

The number $N(t)$ of distinct sites visited by a random walker up to time $t$ is a key property in random walk (RW) theory[1–7] which appears in many physical[8–21], chemical[22,23], and ecological[24] phenomena. This observable quantifies the efficiency of various stochastic exploration processes, such as animal foraging[24] or the trapping of diffusing molecules[1,23]. While the average and, for some examples, the distribution of the number of distinct sites visited, have been determined analytically[5,25–27], this information is far from a complete description. In this work, we show that the waiting time $\tau_n$, defined as the elapsed time between the visit to the $n^{th}$ and the $(n+1)^{st}$ distinct, or new, sites characterizes the exploration dynamics in a more fundamental and comprehensive way (Fig. 1).

In addition to their basic role in characterizing site visitation, the $\tau_n$ are central to phenomena that are controlled by the time between visits to new sites. A class of such models are self-interacting RWs, where a random walker deposits a signal at each visited site that alters the future dynamics of the walker on its next visit to these sites. This self-attracting RW [28–30] has recently been shown to account for real trajectories of living cells[31]. In this model, the probability that the RW jumps to a neighboring site $i$ is proportional to $\exp(-un_i)$, where $u$ is a positive constant, $n_i = 0$ if the site $i$ has never been visited up to time $t$ and $n_i = 1$ otherwise. The analysis of this strongly non-Markovian walk is a difficult problem with few results available in dimension higher than 1. However, we note that its evolution between visits to new sites is described by a

regular RW whose properties are well known. This makes the determination of the statistics of the $\tau_n$ an important first step in the analysis and understanding of these non-Markovian RWs.

The variables $\tau_n$ also underlie starving RWs[32–36], which describe depletion-controlled starvation of a RW forager. In these models, the RW survives only if the time elapsed until a new food-containing site is visited is less than an intrinsic metabolic time $\mathcal{S}$. If the forager collects a unit of resource each time a new site is visited, then in one trajectory, the forager might find resources at an almost regular rate while in another trajectory, the forager might find most of its resources near the end of its wandering. This discrepancy in histories has dramatic effects: the forager survives on the first trajectory but not the latter. To understand this disparity requires knowledge of the random variables $\tau_n$.

Despite their utility and fundamentality, the statistical properties of the $\tau_n$ appear to be mostly unexplored, except for the one-dimensional (1$d$) nearest-neighbor RW. In this special case, the distribution of $\tau_n$ coincides with the classic first-exit probability of a RW from an interval of length $n$, $F_n(\tau)$[37]. We drop the subscript $n$ on $\tau$ henceforth, because the value of $n$ will be evident by context. In the limit $n \to \infty$ with $\tau/n^2$ fixed, $F_n$ has the following basic properties: (i) aging[38]; in general, $F_n$ depends explicitly on $n$, or equivalently, the time elapsed until the visit to the $n^{th}$ new site; (ii) an $n$-independent algebraic decay: $\tau^{-3/2}$ for $1 \ll \tau \ll n^2$, where $n^2$ is the typical time to diffuse across the interval; (iii) an exponential decay for $\tau \gg n^2$; (iv) $F_n$ admits the

[1]Laboratoire de Physique Théorique de la Matière Condensée, CNRS/Sorbonne University, 4 Place Jussieu, 75005 Paris, France. [2]Santa Fe Institute, 1399 Hyde Park Road, Santa Fe, NM 87501, USA. ✉e-mail: benichou@lptmc.jussieu.fr

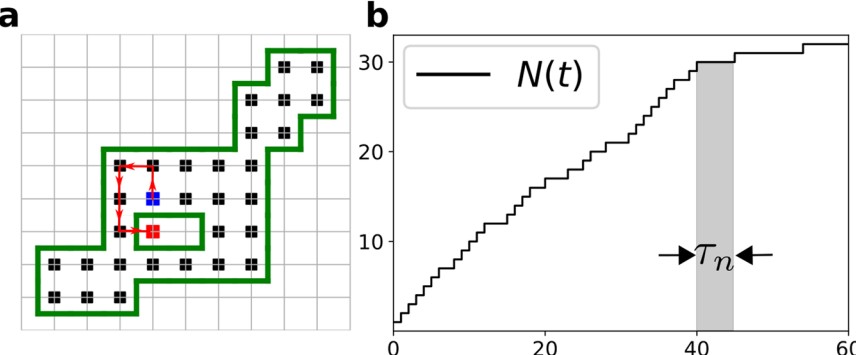

**Fig. 1 | Definition of the random variable $\tau_n$. a** A visited domain (black sites) and its boundary (green line) for a RW on the square lattice. The $n^{\text{th}}$ and $(n+1)^{\text{st}}$ new sites visited are blue and red squares. The red links indicate the intervening RW trajectory. **b** The time intervals $\tau_n$ between increments in $N(t)$, the number of new sites visited at time $t$.

scaling form $F_n(\tau) = n^{-3}\psi(\tau/n^2)$ (see Sec. S1 in the Supplementary Information (SI) for details).

## Results

In this work, we extend these visitation properties to the physically relevant and general situations of higher dimensions and general classes of RWs, including anomalous diffusion. We investigate symmetric Markovian RWs that move in a medium of fractal dimension $d_{\text{f}}$, and whose mean-square displacement is assumed to be given by $\langle \mathbf{r}^2(t) \rangle \propto t^{2/d_{\text{w}}}$, where $d_{\text{w}}$ is the dimension of the walk[39] and $t$ the number of RW steps. We assume in particular the existence of a renewal equation between the propagator and the first-passage time density of the RW[40]. We focus on discrete time and space RWs, for which the number of sites visited at a certain time is clearly defined (see SI S5.C.1 for the extension of our results to Continuous Time Random Walks (CTRWs)). The ratio $\mu = d_{\text{f}}/d_{\text{w}}$ determines whether the RW is recurrent ($\mu < 1$), marginal ($\mu = 1$), or transient ($\mu > 1$). For recurrent and marginal RWs ($\mu \le 1$), the probability to eventually visit any site is one, while for transient RWs ($\mu > 1$), the probability to visit any site is strictly less than one[41,42]. Despite the geometrical complexity of the territory explored after $n$ steps (which typically contains holes, islands[43] and is not spherical[44,45], see Figs. 1 and 2), the distribution of the times $\tau$ between visits to new sites obeys universal statistics that are characterized only by $\mu$, as summarized in Table 1 (up to constant prefactors that are independent of $\tau$, and neglecting algebraic corrections for the two latter regimes).

Fundamental consequences of our results include the following: (i) Finding new sites takes progressively more time for recurrent and marginal RWs; this agrees with simple intuition. This property is quantified by the $n$ dependence of the moments of $\tau_n$. From the entries in Table 1 we find $\langle \tau_n^k \rangle \propto n^{k/\mu-1}$ for recurrent RWs, while $\langle \tau_n \rangle \propto \ln n$ and $\langle \tau_n^k \rangle \propto n^{(k-1)/2}$ for $k > 1$ for marginal RWs. Conversely, transient RWs rarely return to previously visited sites, so that $\langle \tau_n^k \rangle \propto$ const (see the SI Sec. S3.D for the derivation and numerical check). (ii) The statistics of the $\tau_n$ exhibit universal and giant fluctuations for recurrent and marginal RWs, with $\text{Var}(\tau_n)/\langle \tau_n \rangle^2 \propto n$ for recurrent walks and $\text{Var}(\tau_n)/\langle \tau_n \rangle^2 \propto \sqrt{n}/(\ln n)^2$ for marginal walks. In the context of the foraging process mentioned above, this leads to very different life histories of individual foragers. In contrast, $\tau_n$ remains bounded for large $n$ for transient RWs, so that fluctuations remain small. (iii) The early-time regime is independent of $n$. The feature of aging, which originates from the finite size $n$ of the domain visited, arises after a time $t_n$ for recurrent and marginal RWs, and $T_n$ for transient RWs (see Table 1 and below for the definition of these two fundamental time scales). (iv) As shown below, each regime of the exploration dynamics is controlled by specific configurations that are illustrated in Fig. 2. These provide the physical mechanisms that underlie the entries in Table 1. (v) The algebraic decay of $F_n(\tau)$ in the recurrent case should be compared with the simpler problem of a recurrent RW in unbounded space, where the first-passage time distribution to a given target behaves at large times like $F_{\text{target}}(\tau) \propto 1/\tau^{1+\theta}$, with $\theta$ the so-called persistence exponent[46]. Because $\theta = 1 - \mu$ for processes with stationary increments[38], and in particular for Markovian processes, the algebraic decay of $F_n$ in Table 1 can be rewritten as $F_n(\tau) \propto \tau^{-(2-\theta)}$, in sharp contrast with the decay of $F_{\text{target}}(\tau)$. While the two exponents coincide for a simple RW in $1d$ (for which $\theta = 1/2$), the problem here involves the first-exit time statistics from a domain whose complex shape is generated by the RW itself.

We now sketch how to derive these results (see Secs. S2–S3 of the SI for detailed calculations). As an essential step, we first map the visitation problem to an equivalent trapping problem. In our visitation problem, we view unvisited sites as traps for the RW, so that a RW is trapped whenever it leaves the domain of already visited sites. Here, the term trapped does not mean that the RW disappears, but rather, the RW continues its motion but now with the visited domain expanded by the site just visited and the inter-visit time $\tau$ is reset to zero. By this equivalence to trapping, the time $\tau$ between visits to the $n^{\text{th}}$ and $(n+1)^{\text{st}}$ new sites is the same as the probability for the RW to first exit the domain that is comprised of the $n$ already visited sites, or equivalently the domain free of traps. A crucial feature of this equivalence to trapping is that the spatial distribution of traps is continuously updated by the RW trajectory itself. In contrast to the classical trapping problem[47,48], where permanent traps are randomly distributed, here the spatial distribution of traps ages because it depends on $n$. Moreover, successive traps are spatially correlated, with correlations generated by the RW trajectory.

These two key points are accounted for by the distribution $Q_n(r)$ of the radius of the largest spherical region that is free of traps after $n$ sites have been visited. We show in Sec. S2.D of the SI that this distribution assumes the scaling form $Q_n(r) \simeq \rho_n^{-1} \exp[-a(r/\rho_n)^{d_{\text{f}}}]$, where $a$ is independent of $n$ and $r$ and the characteristic length $\rho_n$ provides the typical scale of this radius $r$. Furthermore, the $n$ dependence of $\rho_n$, which quantifies both aging and correlations between traps, is determined by whether the exploration is recurrent or transient. Specifically, we find $\rho_n = n^{1/d_{\text{f}}}$ for $\mu < 1$, $\rho_n = n^{1/2d_{\text{f}}}$ for $\mu = 1$ and $\rho_n$ of the order of one, up to logarithmic corrections for $\mu > 1$ (see Sec. S2 in the SI). A striking feature of these behaviors is that the exponent changes discontinuously when $\mu$ passes through 1.

The corresponding time scales $t_n = \rho_n^{d_{\text{w}}}$ and $T_n$ delineate the three regimes of scaling behaviors summarized in Table 1 and Fig. 2: (i) a short-time algebraic regime ($1 \ll \tau \ll t_n$), (ii) an intermediate-time stretched exponential regime ($t_n \ll \tau \ll T_n$), and (iii) a long-time exponential regime ($T_n \ll \tau$). Here $T_n$ is defined as the time at which the radius of the trap-free region $r^*(\tau)$ that controls the dynamics takes its

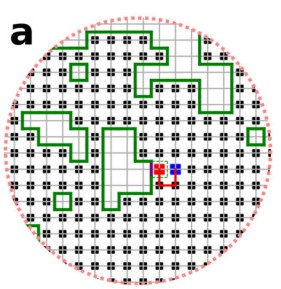
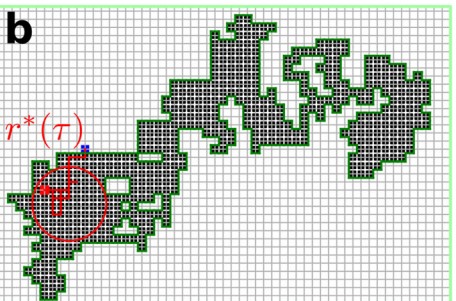
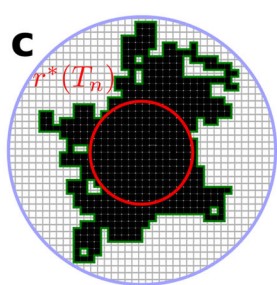

**Fig. 2 | The three temporal regimes of the exploration dynamics, as illustrated by a RW on a square lattice.** Each panel shows the corresponding different controlling configurations when $n = 500$ distinct sites have been visited. The $n^{th}$ and $(n+1)^{st}$ visited site are shown in red and blue, respectively (**a** and **b**). **a** Early time: the visited domain (black squares within the green boundary) is effectively infinite (at the scale of the trajectory of the RW during the time $\tau_n$). **b** Intermediate time: the exit time probability from the visited domain is governed by atypically large trap-free regions of radius $r^*(\tau) \sim \rho_n (\tau/t_n)^{1/(d_f + d_w)}$. **c** Long time: the exit time probability is determined by atypically large trap-free regions of radius $r^*(T_n) \sim n^{1/d_f}$.

maximal possible value of $r_{\max} = n^{1/d_f}$ (see Fig. 2c and the discussion below Eq. (4)). We do not characterize the early time regime $\tau = O(1)$ which depends on details of the model: we are only interested in universal features.

## Algebraic regime

Here, the distribution of $\tau$ has a universal algebraic decay whose origin stems from two essential features: (i) The RW just visited a new site so that the RW starts from the interface between traps and visited sites when the clock for the next $\tau$ begins. (ii) The region already visited by the RW is sufficiently large so that we can treat the region as effectively infinite (Fig. 2a) and thereby approximate $F_n(\tau)$ by $F_\infty(\tau)$.

The first-return time distribution to this set of traps on the interface is determined by the renewal equation[40,49,50] that links the probability $P_{\mathrm{trap}}(t)$ to be at a trap at time $t$ and the distribution of first arrival times $F_\infty(\tau)$ to a trap at time $\tau$,

$$P_{\mathrm{trap}}(t) = \delta(t) + \int_0^t F_\infty(\tau) P_{\mathrm{trap}}(t - \tau)\, d\tau. \qquad (1)$$

This equation expresses the partitioning of the total RW path to the interface into a first-passage path to the interface over a time $\tau$ and a return path to the interface over the remaining time $t - \tau$; here we use a continuous-time formulation for simplicity. In this mean-field type equation (detailed in SI Sec. S3.A.*1* and *2*, and supported by numerical simulations given below and an alternative derivation for the exponent of the algebraic decay given in Sec. S3.A.*3* in the SI), we treat the set of traps collectively, which amounts to neglecting correlations between the return time and the location of the traps on the interface.

Next, we estimate $P_{\mathrm{trap}}(t)$ by using the fact that the RW is almost uniformly distributed in a sphere of radius $r(t) \propto t^{1/d_w}$ at time $t$. The number of traps within this sphere is given by $r(t)^{d_T}$. Here $d_T$ is the fractal dimension of the interface between visited and non-visited sites; as shown in the SI Sec. S3.A.*2*, $d_T = 2d_f - d_w$. Finally, we obtain the fraction of traps within this sphere and thereby $P_{\mathrm{trap}}(t)$:

$$P_{\mathrm{trap}}(t) \propto \frac{\text{Number of traps}}{\text{Number of sites}} \propto \frac{r(t)^{d_T}}{r(t)^{d_f}} \propto t^{\mu - 1}. \qquad (2)$$

Based on (2), we solve Eq. (1) in the Laplace domain and invert this solution to obtain the algebraic decay $F_\infty(\tau) \simeq A\tau^{-1-\mu}$ in Table 1 in the early-time regime for recurrent and marginal RWs (this derivation is given in Sec. S3.A in the SI, including exact and approximate expressions for the amplitude $A$ for marginal and recurrent RWs, respectively).

In the transient case, the RW is always close to a non-visited site by the very nature of transience. Consequently, the time scale $t_n$ is of order one and the algebraic regime does not exist.

## Intermediate- and long-time regimes

If the RW survives beyond the early-time regime, it can now be considered to start from within the interior of the domain of visited sites. In analogy with the classical trapping problem, a lower bound for the survival probability of the RW, $S_n(\tau)$, is just the probability for the RW to remain within this domain. This lower bound is controlled by the rare configurations of large spherical trap-free regions in which the RW starts at the center of this sphere, whose radius distribution $Q_n(r)$ was given above.

We develop a large-deviation approach, in which this lower bound is given by the probability $q_n$ for the RW to first survive up to the first crossover time $t_n$, multiplied by the probability for the RW to remain inside a spherical trap-free domain over a time $\tau$. The quantity $q_n$ is given by $\int_{t_n}^\infty F_\infty(\tau)d\tau$, which scales as $1/t_n^\mu$ if $\mu \leq 1$, and is of order one if $\mu > 1$. The probability for the RW to remain inside a spherical domain of radius $r$ over a time $\tau$ asymptotically scales as $\exp(-b\,\tau/r^{d_w})$, where $b$ is a constant[39]. As stated above, the probability to find a spherical trap-free region of radius $r$ is given by $Q_n(r) \simeq \rho_n^{-1} \exp[-a(r/\rho_n)^{d_f}]$. Summing over all radii up to the largest possible value $r_{\max} = n^{1/d_f}$, we obtain the lower bound

$$S_n(\tau) \geq \frac{q_n}{\rho_n} \int_0^{n^{1/d_f}} \exp\left[-b\tau/r^{d_w} - a(r/\rho_n)^{d_f}\right] dr, \qquad (3)$$

where $a$ and $b$ are constants. Using Laplace's method by making the change of variable $r = \rho\tau^{1/(d_f + d_w)}$, we obtain (ignoring algebraic prefactors in $n$ and $\tau$),

$$\begin{aligned} S_n(\tau) &\geq \int_0^{n^{1/d_f}} \exp\left[-\tau^{\mu/(1+\mu)}\left(b/\rho^{d_w} + a(\rho/\rho_n)^{d_f}\right)\right] d\rho \\ &\geq \exp\left[-\tau^{\mu/(1+\mu)}\left(b/\rho^{*d_w} + a(\rho^*/\rho_n)^{d_f}\right)\right], \end{aligned} \qquad (4)$$

where the function $b/\rho^{d_w} + a(\rho/\rho_n)^{d_f}$ reaches its minimum at $\rho = \rho^*$. The lower bound (4) for $\tau \gg 1$ is controlled by trap-free regions of radius $r^*(\tau) = \rho^* \tau^{1/(d_w + d_f)} \sim \rho_n^{d_f/(d_w + d_f)} \tau^{1/(d_w + d_f)}$ (see SI Sec. S3.B for details). Using $t_n = \rho_n^{d_w}$, this optimal radius is then $r^*(\tau) \sim \rho_n(\tau/t_n)^{1/(d_w + d_f)}$. For $\tau \gg t_n$, we have $r^*(\tau) \gg \rho_n$. Since $\rho_n$ determines the typical radius of the largest spherical region free of traps, the configurations that control the long-time dynamics (as illustrated in Fig. 2b, c) are atypically large, and become more so as $\tau$ increases. Thus the survival probability in this long-time regime is determined by a compromise between the scarceness of large trap-free domains and the long exit times from such domains. Finally, we obtain

**Table 1 | Summary of the time dependence of $F_n(\tau)$ for the three classes of RWs—recurrent, marginal, and transient**

|  | $t_n$ | $T_n$ | $1 \ll \tau \ll t_n$ | $t_n \ll \tau \ll T_n$ | $T_n \ll \tau$ |
|---|---|---|---|---|---|
| $\mu < 1$ [recurrent] | $n^{1/\mu}$ | $n^{1/\mu}$ | $\tau^{-(1+\mu)} \equiv \tau^{-(2-\theta)}$ |  | $\exp[-\mathrm{const}\,\tau/n^{1/\mu}]$ |
| $\mu = 1$ [marginal] | $\sqrt{n}$ | $n^{3/2}$ | $\tau^{-(1+\mu)} \equiv \tau^{-(2-\theta)}$ | $\exp\left[-\mathrm{const}\left(\tau/t_n\right)^{\mu/(1+\mu)}\right]$ | $\exp[-\mathrm{const}\,\tau/n^{1/\mu}]$ |
| $\mu > 1$ [transient] | $1$ | $n^{(\mu+1)/\mu}$ |  | $\exp\left[-\mathrm{const}\left(\tau/t_n\right)^{\mu/(1+\mu)}\right]$ | $\exp[-\mathrm{const}\,\tau/n^{1/\mu}]$ |

The constants are independent of $n$ and $\tau$. The crossover times $t_n$ and $T_n$ are given up to logarithmic prefactors. The time regimes identified in the last three columns are the same as the ones presented in Fig. 2. The persistence exponent $\theta$ is here given by $\theta = 1 - \mu$, see text.

$F_n(\tau) = -\mathrm{d}S_n(\tau)/\mathrm{d}\tau \sim \exp[-\mathrm{const}(\tau/t_n)^{\mu/(1+\mu)}]$. As in the classic trapping problem [1,3,40], we expect that this lower bound for the survival probability will have the same time dependence as the survival probability itself.

This stretched exponential decay holds as long as the optimal radius is smaller than the maximal value $r_{\max}$. The point at which this inequality no longer holds defines a second crossover time $T_n$ by $r^*(T_n) = n^{1/d_f}$. Beyond this time, the evaluation of the integral in Eq. (4) now leads to an exponential decay of $F_n$ (Table 1).

Finally, note that the full time dependence of $F_n(\tau)$ has a particularly simple form for recurrent RWs. In this case, the intermediate stretched exponential regime does not exist because $t_n$ and $T_n$ both have the same $n$ dependence. In fact, the short- and long-time limits of $F_n(\tau)$ can be synthesized into the scaling form (as explained in Sec. S3.C of the SI)

$$F_n(\tau) = \frac{1}{n^{1+1/\mu}} \psi\left(\frac{\tau}{n^{1/\mu}}\right), \qquad (5)$$

with $\psi$ a scaling function.

We confirm the validity of our analytical results by comparing them to numerical simulations of paradigmatic examples of RWs that embody the different cases in Table 1. The recurrent case ($\mu < 1$) is illustrated in Fig. 3a–c for diverse processes: superdiffusive Lévy flights in $1d$, in which the distribution of jump lengths is fat-tailed, $p(\ell) \propto \ell^{-1-\alpha}$, with $\alpha \in ]1,2[$; subdiffusive RWs on deterministic fractals with and without loops, respectively represented by the Sierpinski gasket and the T-tree (see Sec. S4.A of SI for the definition of the T-tree and the simulation results); subdiffusive RWs on disordered systems, as represented by a critical percolation cluster on a square lattice. Our simulations confirm the scaling form of $F_n(\tau)$ given in Eq. (5), as well as its algebraic ($X \equiv \tau/t_n < 1$) and exponential ($X > 1$) decays at respectively short and long times.

The marginal case ($\mu = 1$) is illustrated by $1d$ Lévy flights of parameter $\alpha = 1$, persistent and simple RWs on the 2-dimensional square lattice (Fig. 3d–f respectively). The data collapse when plotted versus the scaling variable $\tau/\sqrt{n}$; this confirms that the crossover time $t_n$ scales as $t_n \propto \sqrt{n}$. Figure 3d and e clearly show the expected algebraic decay $\tau^{-2}$ at short times (dashed line). Figure 3f validates the stretched exponential form of $F_n(\tau)$ at intermediate times, as well as the exponential decrease at long times and the scaling of $T_n = n^{3/2}$.

The transient case ($\mu > 1$) is illustrated by RWs on hypercubic lattices (see Fig. 3g for the $2d$ Lévy flights of parameter $\alpha = 1$, Fig. 3h for a persistent RW and Fig. 3i for a nearest neighbour RW in $3d$, as well as Sec. S4.C.5 in the SI for higher dimensions and Sec. S4.C.6 for transient Lévy flights). Figure 3i confirms the stretched exponential temporal decay for intermediate times, the scaling of the crossover time $T_n = n^{1/\mu+1}$, and the long-time exponential decay of $F_n(\tau)$ for transient RWs. The numerically challenging task of observing the stretched exponential decay followed by the exponential decay that originates from rare, trap-free regions, was achieved by relying on Monte Carlo simulations coupled with an exact enumeration technique (see Sec. S4.C of the SI for details). We note that in Fig. 3g and h, the distribution is independent of $n$ for the values of $X \equiv \tau$ represented,

and $Y = -\left(\ln F_n(\tau)\right)/\tau^{\mu/(1+\mu)}$ reaches a plateau. It further confirms the stretched exponential regime and the absence of the algebraic regime ($t_n = 1$).

Overall, we find excellent agreement between our analytical predictions and numerical simulations. The diverse nature of these examples also demonstrates the wide range of applicability of our theoretical approach.

We can extend our approach to treat the dynamics of other basic observables that characterize the support of RWs. Following [51,52] two classes of observables can be defined: boundary and bulk. Boundary observables involve both visited and unvisited sites, such as the perimeter $P(t)$ of the visited domain or the number of islands $I(t)$ enclosed in the support of the RW trajectory; note that these variables can both increase and decrease with time. We show, for example, in Sec. S5.A of the SI, that the corresponding distribution of the times between successive increases in a boundary observable $\Sigma$ again has an early-time algebraic decay, $F_\Sigma(\tau) \propto \tau^{-2\mu}$ for $\mu < 1$, and $F_\Sigma(\tau) \propto \ln \tau/\tau^2$ for $\mu = 1$. These behaviors are illustrated in Fig. 4a–c. Bulk observables involve only visited sites, such as the number of dimers[51], $k$-mers, and $k \times k$ squares in $2d$. We show in Sec. S5.A of the SI that the dynamics of bulk variables is the same as that for the number of distinct sites visited.

## Discussion

In addition to providing asymptotic expressions for the $\tau_n$ distribution and their extension to basic observables characterizing the support of RWs, our results open new avenues in several directions. First, they allow us to revisit the old question of the number $N(t)$ of distinct sites visited at time $t$. Indeed, our theoretical approach for the set of inter-visit times $\tau$ represents a start towards determining multiple-time visitation correlations for general RWs, quantities that have remained inaccessible this far. These multiple-time correlations are crucial to fully characterize the stochastic process $\{N(t)\}$, the number of sites visited at every single time. However, they have been studied only for the special case of $1d$ nearest-neighbor RWs [27,53]. Using our formalism we can further compute temporal correlations of $\{N(t)\}$ for compact Lévy flights in $1d$ with $1/\mu = \alpha > 1$ (which do leave holes in their trajectories). We compute the scaling with time of the two-time covariance of the number of distinct sites visited,

$$\mathrm{Cov}[N(t_1), N(t_2)] \equiv \langle N(t_1)N(t_2) \rangle - \langle N(t_1) \rangle \langle N(t_2) \rangle.$$

We obtain in the limit $1 \ll t_1 \ll t_2$ (see Sec. S5.B of the SI for a numerical check of the derivation of Eq. (6) and its numerical confirmation which can also be seen in Fig. 4d),

$$\mathrm{Cov}[N(t_1), N(t_2)] \propto t_1^\mu t_2^\mu \frac{t_1}{t_2}. \qquad (6)$$

This result can be further extended to $k$-time correlation functions (see the numerical confirmation for $k = 4$ in Fig. 4e),

$$\langle (N(t_1) - \langle N(t_1) \rangle) \ldots (N(t_k) - \langle N(t_k) \rangle) \rangle \propto t_1^\mu \ldots t_k^\mu \frac{t_1}{t_k}. \qquad (7)$$

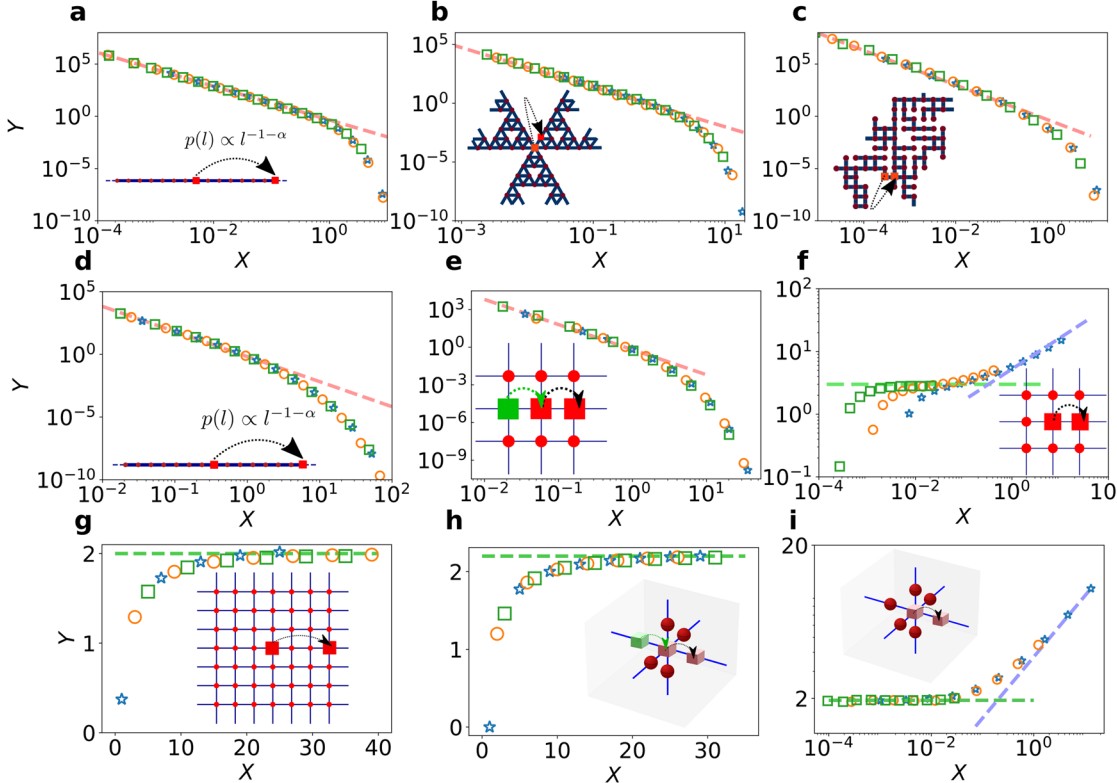

**Fig. 3 | Universal distribution of the time between visits to new sites for RWs.**
**Recurrent RWs** ($\mu < 1$). Shown is the scaled distribution $Y \equiv \theta_n^{1+\mu} F_n(\tau)$ versus $X \equiv \tau/\theta_n$ for $n = 100$, 500, and 1000. Here $\theta_n \sim n^{1/\mu}$ is the decay rate of the exponential in $F_n(\tau) \sim \exp(-\tau/\theta_n)$. The red dashed lines indicate the algebraic decay $A(\mu)\tau^{-1-\mu}$ ($A(\mu)$ defined in SI Sec. S3.A.4). **a** $1d$ Lévy flights with index $\alpha = 1/\mu = \ln 6/\ln 3$.
**b** Subdiffusion on a Sierpinski gasket ($\mu = \ln 3/\ln 5$, scaling of $\theta_n$ with $n$ shown in SI Sec. S4.A). **c** Subdiffusion on a $2d$ critical percolation cluster ($\mu \approx 0.659$). **Marginally recurrent RWs** ($\mu = 1$). **d e** Marginal RWs ($\mu = 1$) at early times. Shown is the scaled distribution $Y \equiv n F_n(\tau)$ versus $X \equiv \tau/\sqrt{n}$ for **d** $1d$ Lévy flights of index $\alpha = 1$ for $n = 800$, 1600, and 3200, **e** persistent RWs in $2d$ where the probability to continue in the same direction is $p = 0.3$ for $n = 800$, 1600 and 3200. The red dashed line represent the algebraic decay $A\tau^{-2}$ ($A$ given in SI Sec. S3.A.4). **f** Marginal RWs at intermediate and long times. Shown is the scaled distribution $Y \equiv$

$(-\ln n F_n(\tau))/\sqrt{\tau/\sqrt{n}}$ versus $X \equiv \tau/n^{3/2}$ for simple RWs on a $2d$ square lattice for $n = 200$, 800 and 3200. The green and blue dashed lines represent the stretched exponential and the exponential regimes, respectively. **Transient RWs** ($\mu > 1$). Shown is the scaled distribution $Y \equiv (-\ln F_n(\tau))/\tau^{\mu/(1+\mu)}$ for **g** Lévy flights of parameter $\alpha = 1$ in $2d$, for $n = 400$, 800, 1600 and $X \equiv \tau$, **h** persistent RWs in $3d$ where the probability to continue in the same direction is $p = 0.25$ for $n = 200$, 800, 3200 and $X \equiv \tau$, **i** simple RWs on cubic lattice, for $n = 200$, 400, 500 and $X \equiv \tau/n^{1+1/\mu}$. The green and blue dashed lines represent the stretched exponential and the exponential regimes, respectively. For all panels, blue stars, orange circles and green squares correspond to increasing values of $n$. The insets indicate the jump processes. Red squares are the initial and arriving positions of the walker. The green squares represent the prior position of the walker.

To obtain these results, we rely on the assumption that for any values of the number of distinct sites visited $n_1$ and $n_2$ holds

$$\mathrm{Cov}\left[\sum_{k=0}^{n_1-1}\tau_k, \sum_{k=n_1}^{n_2-1}\tau_k\right] = O\left(n_1^{2/\mu}\right), \tag{8}$$

which is indeed verified for $1d$ Lévy flights (see SI Sec. S5 B). In addition to the case of $1d$ Lévy flights, where Eq. (8) is satisfied, Eqs. (6) and (7) provide in fact lower bounds on the correlation functions for recurrent RWs (see SI Sec. S5.B for numerical checks),

$$\langle (N(t_1) - \langle N(t_1)\rangle)\dots(N(t_k) - \langle N(t_k)\rangle)\rangle \geq t_1^\mu \dots t_k^\mu \frac{t_1}{t_k}. \tag{9}$$

This lower bound is algebraically decreasing in $t_k$. The salient feature of these results is that temporal correlations in multiple-time distributions of recurrent RWs, such as those in Eq. (6), have a long memory.

Second, the distribution of $\tau_n$ allows us to provide a quantitative answer to the question raised in the introduction regarding the disparity in life histories of foragers that starve if they do not eat after $\mathcal{S}$ steps. While in $1d$, the mean starvation time is known to increase linearly with $\mathcal{S}$ (at large $\mathcal{S}$), the corresponding question in

$2d$, which is relevant to most applications of foraging, is open. We now show, by relying on the results introduced in this paper, that the mean number of sites visited and consequently the starvation time in $2d$ increases quadratically with $\mathcal{S}$ (up to logarithmic corrections). We start with the observation that, knowing that $n$ sites have been visited, the probability to starve is given by the probability that the time $\tau_n$ to visit a new site is larger than the metabolic time $\mathcal{S}$, $\mathbb{P}(\tau_n > \mathcal{S}) = \sum_{\tau > \mathcal{S}} F_n(\tau)$. Using Table 1, we have that for $t_n = \sqrt{n} < \mathcal{S}$, the probability to starve is stretched exponentially small (up to algebraic prefactors), $\mathbb{P}(\tau_n > \mathcal{S}) \approx \exp[-\sqrt{\mathcal{S}/t_n}]$. The desert (domain witout food) formed by the set of visited sites is too small to prevent the RW from finding new sites: the RW visits $\mathcal{S}^2$ sites in total in this first regime. However, for $t_n = \sqrt{n} > \mathcal{S}$, the probability for the RW to starve before finding a new site is large, as it is given by the tail of an algebraic distribution $\mathbb{P}(\tau_n > \mathcal{S}) \propto 1/\mathcal{S}$. Consequently, the number of sites visited in this regime is negligible compared to the first one. Thus, the number of sites visited at starvation is given, up to log corrections, by $n = \mathcal{S}^2$ and the lifetime by $\sum_{k=1}^{\mathcal{S}^2}\langle\tau_k\rangle \sim \mathcal{S}^2$. This result is confirmed numerically in Fig. 4f. This resolves the open question of the lifetime of $2d$ starving RWs[32–36].

Finally, the generality of our results opens the question of extending them to the challenging situation of non-Markovian

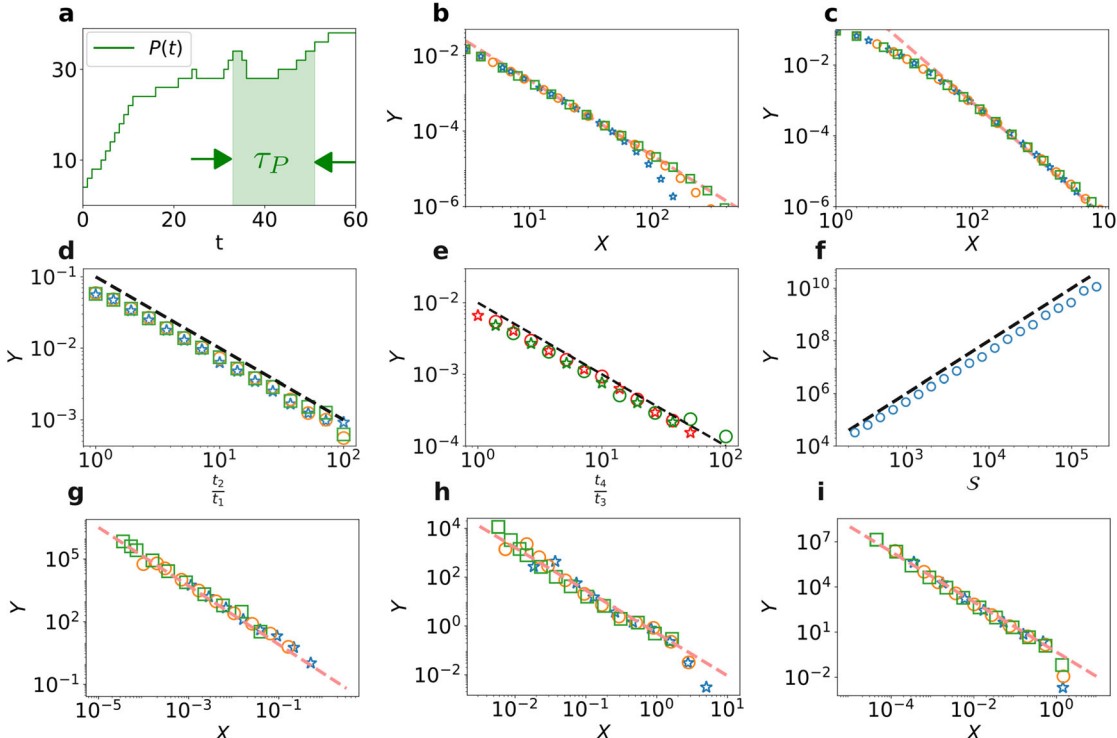

**Fig. 4 | Extensions and applications of the time between visits to new sites for RWs. Boundary observables for recurrent and marginal RWs:** The perimeter of the visited domain and the number of islands enclosed in the support. **a** The elapsed time $\tau_P$ for successive increments of the time dependence of the perimeter $P(t)$ of the visited domain. **b** Distribution $F_P(\tau)$ of the time elapsed $\tau_P$ between the first observations of a domain perimeter of length $P$ and $P+2$ for simple RWs on the square lattice. **c** Distribution $F_I(\tau)$ of the elapsed time $\tau_I$ between the first occurrence of $I$ and $I+1$ islands for Lévy flights of index $\alpha = 1.2$. Plotted in **b** and **c** are the scaled distributions $Y \equiv F_P(\tau)/(\ln 8\tau)$ and $Y \equiv F_I(\tau)$ versus $X = \tau$. The red dashed lines have slope $-2\mu$. The data are for $P, I = 50, 100,$ and $200$ (respectively blue stars, orange circles and green squares). **Multiple-time covariances and starving RWs. d** $Y = \text{Cov}[N(t_1), N(t_2)]/(\langle N(t_1)\rangle\langle N(t_2)\rangle)$ for Lévy flights of parameter $\alpha = 1.5$, and we compare $Y$ to $\frac{t_1}{t_2}$ (dashed line). The stars, circles, and squares indicate data for

$t_1 = 10, t_1 = 100,$ and $t_1 = 1000$. **e** $Y = \frac{t_3}{t_1}\langle(N(t_1) - \langle N(t_1)\rangle)(N(t_2) - \langle N(t_2)\rangle)(N(t_3) - \langle N(t_3)\rangle)(N(t_4) - \langle N(t_4)\rangle)\rangle/(\langle N(t_1)\rangle\langle N(t_2)\rangle\langle N(t_3)\rangle\langle N(t_4)\rangle)$, for Lévy flights of parameter $\alpha = 1.3$. We compare $Y$ to the dashed line proportional to $\frac{t_3}{t_4}$. Data in red and green indicate $t_1 = 10$ and $t_1 = 100$. Stars indicate $t_2 = 2t_1$ and circles indicate $t_2 = 4t_1$. We take $t_3 = 4t_2$. **f** Lifetime at starvation. Blue circles show the mean lifetime versus the metabolic time $S$. The dashed line is proportional to $S^2$. **Non-Markovian examples.** Rescaled distribution $Y = F_n(\tau)n^{1+1/\mu}$ versus $X = \tau/n^{1/\mu}$ for **g** Fractional Brownian motion with parameter $1/H = 1/0.4 = \mu = 1 - \theta$ ($n = 20, 40$ and $80$) **h** Fractional Brownian motion with parameter $1/H = 1/0.75 = \mu = 1 - \theta$ ($n = 20, 40$ and $80$), **i** True Self Avoiding Walks $\mu = 2/3 = 1 - \theta$ ($n = 200, 400$ and $800$). For the last three panels, increasing values of $n$ are represented successively by blue stars, orange circles and green squares, and the dashed line is proportional to $X^{-(2-\theta)}$.

processes, which is a priori not covered by our approach. However, we argue in SI Sec S5.C that our results concerning the recurrent case can be extended to non-Markovian processes. The agreement with numerical simulations of highly non-Markovian processes such as the Fractional Brownian Motion[54] (in the sub- and super-diffusive cases) and the True Self Avoiding Walk[55] (see SI Sec S5.C.4 for definition) is displayed in Fig. 4 (g–i respectively). We point out again that this behavior $F_n(\tau) \propto \tau^{-(2-\theta)}$ is in sharp contrast to the usual decay of the first-passage probability to a target $F_{\text{target}}(\tau) \propto \tau^{-(1+\theta)}$. This difference originates both from the complex geometry of the support of the RW and potential memory, which, remarkably, are universally accounted for by our results.

We have shown that the times between successive visits to new sites are a fundamental and useful characterization of the territory explored by a RW. We identified three temporal regimes for the behavior of these inter-visit time distributions, as well as the physical mechanisms that underlie these different regimes. In addition to their fundamental nature, these inter-visit times satisfy strikingly universal statistics, in spite of the geometrical complexity of the support of the underlying RW processes. The elucidation of these inter-visit times represents a promising research avenue to discover many more aspects of the intriguing exploration dynamics of RWs, as shown by the first applications provided here in the case of non-Markovian processes.

## Methods

Analytical results are verified using simulations of different RW models:

### Numerical simulations of recurrent and marginal RWs

- Lévy flights in $1d$ with $\alpha \in [1,2[$, where the jump length is drawn from $p(s) = 1/[2\zeta(1+\alpha)|s|^{1+\alpha}]$. Intermediary sites between initial and final positions of the jump are not visited.
- Nearest-neighbour RWs on the Sierpinski gasket. The gasket is unbounded, and each neighbouring site is chosen with equal probability. Each RW starts at the central site (red square on Fig. 3b).
- Nearest-neighbour RWs on the T tree. The T tree is generated up to generation 9, and then we perform a RW starting at the central site. Each neighbouring site is chosen with equal probability.
- Nearest-neighbour RWs on critical percolation clusters. The clusters are constructed from a $1000 \times 1000$ periodic square lattice, from which half of the bonds were randomly removed and then the largest cluster was selected. We start from a site chosen uniformly on the cluster. Each neighbouring site is chosen with equal probability.
- Nearest-neighbour RWs on the $2d$ lattice, persistent and not persistent. For persistent RW, the probability to do the same

step as the previous one is larger than 1/4, while the probability to go in any other direction is taken uniformly among the 3 directions left.

We perform the RWs to get the domain $\mathbf{r}_n$ of $n$ distinct visited sites. To obtain the time $\tau_n$ to visit a new site, we use the exact enumeration method based on the adjacency matrix $M(\mathbf{r}_n)$ of the visited domain. The $\theta_n$, based on which the rescaled data lead to Fig. 3, are obtained by measuring the slope of the exponential decrease at large times of the statistics of $\tau_n$.

### Numerical simulations of transient RWs

In addition to the exact enumeration used to obtain the exit time statistics from the visited domain $\mathbf{r}_n$, we rely on a Monte Carlo Markov Chain generation of $\mathbf{r}_n$ on hypercubic lattices $d = 3, 4, 5$ and $6$ (we generate the visited domains in the same way as for recurrent RWs for the persistent RW in $3d$ or transient Lévy flights). Using the observation that the average exit time is proportional to the surface of the visited domain, we bias the generation of these domains towards states of small surfaces. The bias is generated by a Wang-Landau procedure, in order to obtain a uniform probability on the surface of the visited domain, resulting in an increased probability of the small surface states.

### Numerical simulations of non-Markovian RWs

For the True Self Avoiding Random Walk on the $1d$ line, we record the number of visits $C_i$ of site $i$. The probability to jump to the site on the right is given by $\exp(-C_{i+1})/(\exp(-C_{i+1}) + \exp(-C_{i-1}))$, otherwise the RW jumps on the left. For the fractional Brownian motion (fBM), we use the module fbm[56] of python based on Hosking's method[57]. We discretize the line in intervals of size one, and consider that an interval has been visited when the RW enters it for the first time. $\tau_n$ is the time elapsed between visit of the $n^{\text{th}}$ interval and the new $(n+1)^{\text{st}}$ interval.

## Data availability

The data generated in this study have been deposited in a GitHub repository[58] located at: https://github.com/LeoReg/Universal ExplorationDynamics.git.

## Code availability

The code used to generate the data presented in this study have been deposited in a GitHub repository[58] located at: https://github.com/LeoReg/UniversalExplorationDynamics.git.

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

## Acknowledgements
S.R. gratefully acknowledges partial financial support from NSF grant DMR-1910736. We thank A.K. Hartmann, P. Viot and J. Klinger for useful discussions.

## Author contributions
O.B., L.R. and M.D. contributed to analytical calculations. L.R., M.D. and S.R. performed numerical simulations. All the authors wrote the manuscript. O.B. conceived the research.

## Competing interests
The authors declare no competing interests.
