## [Peer Review File · Nature Communications]

Universal exploration dynamics of random walksREVIEWER COMMENTS

Reviewer #1 (Remarks to the Author):

In the present manuscript the authors explore in a very general form the dynamics of the time between visits to newly visited sites for a large class of random walk processes. They derive different regimes for recurrent, marginal and transient walks, and provide corresponding scalings of the intervisit time distributions. They do this by mapping this problem into a trapping problem with distributed traps and introducing several assumptions in order to deal with the temporal and spatial correlations present in the shape of the walker's visited/empty regions. Also, numerical results are presented for a variety of random walk models/simulations in order to support the validity of the different regimes reported analytically.

In my opinion, the whole work seems to be technically correct and sound, and it is written in a clear and precise way. My main concerns are about the real relevance and implications of the results presented. In particular, I do not feel able to assess properly that relevance until the authors have clarified two main points:

- i) The numerical results reported confirm that the scalings found for the different regimes provide a good fit to a large class of random walks. However, in my opinion this does not necessarily mean that all the previous assumptions made to obtain such results are completely valid. It could be that such scaling emerge somehow even if those hypothesis are only partially valid (or even wrong). So, in my opinion the authors should provide a more exhaustive numerical analysis checking directly (not indirectly) the validity of those assumptions. I am particularly intrigued about the assumption made in the derivations in Section S2 that "behavior of all nodes in the ball of radius $r^{\{\gamma\}}$ is equivalent, and the exploration of these nodes are disjoint events". I do not find this intuitive at all, so I would urge the authors to provide evidences that such assumption is really sound. The same would apply to the assumption of independence between 'tau' variables in Section S5.B, which is only checked indirectly.
- ii) Regarding the real relevance and impact that studying the distribution of elapsed times between visits may have, the arguments provided in the introduction are not very convincing. While it is true that times between new visits are more fundamental magnitudes than the mean number of visited sites or other similar measures of search exploration, it is not clear whether this line of research has the potential to generate a shift in the interest of workers in the field. In my opinion, the arguments provided in the last part of the main text (about the opportunities to explore multiple-time correlations between visits) are promising at this respect, but they are only barely sketched in the manuscript. If the authors really believe that this is a real possibility for future research, they should provide more tangible arguments than those simply based on the correlations of $N(t)$ for Lévy flights (again, I have my concerns about the assumptions made in Section S5.B) to support it.

Minor point:

- iii) The authors provide a short discussion about the dispersion of tau values in page 2 of the main text, but very few is shown about this. For the sake of completeness this is an issue that should be explicitly shown, at least as a part of the Supplementary Material.

In conclusion, while the work might have some potential interest and relevance to a broad audience as that of Nat. Comm., I cannot recommend publication of the manuscript in its present form.

Reviewer #2 (Remarks to the Author):

The manuscript by Regnier et al studies the statistics of time intervals between visits of two consequent previously unvisited sites by a random walk. Results of the analytical evaluation (mostly asymptotic analysis of various regimes) are validated with numerical simulations. A a broad range of examples of random walks from 1D Levy flights to fractal structures on 2D lattice and hypercubic

lattices, is considered. The paper is clearly written, with only most relevant key results retained in the main text of the manuscript while all derivations and technical steps are delegated to the supplemental material.

In the context of this manuscript being submitted to Nature Communications I have the following critical comments. In essence, the manuscript is a meticulous analysis of different regimes of the interval statistics for different types of random walks (as defined by the parameter μ). I totally enjoyed reading it as a person working in this field and learned new things. While I also totally agree with the fundamental mathematical nature of the problem studied, the way it is presented does not convince that this problem is of broad interest as implied by the Nature Communications standards. The manuscript is more of a format of a specialised journal, like PRE or Journal of Statistical Physics. The authors claim that the quantity analysed is “fundamental and remarkably useful” but why is it so is never actually shown and discussed. I also actually doubt that one could claim anything like that objectively. Motivation for the analysis of this quantity is also not convincing, which is similar to the argument that the statistics of the distinct sites visited by a random walk is important and thus also the time intervals between the two most recent visited new sites. No practical examples of the importance and/or relevance of the presented results were provided (except for the rather vague discussion on cells with metabolic clocks). Here, the fact that RWs are ubiquitous across disciplines, does not work as a strong argument for the broad interest in the particular quantity studied in this work. To summarise, I do not think that this work is of general importance and interest to the broad readership of Nature Communication and therefore can not recommend it for publication.

Reviewer #3 (Remarks to the Author):

Title: Universal exploration dynamics of random walks

Authors: Léo Régnier, Maxim Dolgushev, S. Redner, and Olivier Bénichou

The authors introduce and explore an, as far as I can tell, novel quantity to characterise the spreading dynamics of random walks, which is the probability distribution of the elapsed time between visits of a walker to distinct (new) sites. It is intuitively evident that this quantity is highly non-trivial and non-stationary, as it characterises the stochastic process in terms of the ‘pattern’ of different sites visited in time, which in turn changes profoundly with time. They calculate this distribution for symmetric random walks by a combination of analytical methods, as explained in detail in the Supplemental Material. These approximate analytical results are supported by computer simulations for a variety of selected random walks models. The key results are summarised in table 1, which gives the asymptotic time dependence of the ‘exploration (or visitation) time distribution’ (as one may call it) for different regimes of time, and for different fundamental classes of random walks. It is claimed that these results are universally valid for symmetric random walks.

I find the article very well written. The main ideas are clearly summarised in the main text while all technical details are shifted into the Supplement, which indeed shows a lot of knowledge and skills. Figure 1 nicely represents the main idea of this approach, Fig.2 together with the table yields a concise summary of the main results including the underlying physics. The essence of this work is thus well accessible to a general physics readership.

I like the main idea underlying this research, which is to study the ‘exploration time distribution’. This quantity indeed goes beyond more conventional approaches characterising random walks by position distributions, associated mean square displacements, or first passage time distributions. Obviously there is a lot of physical content in this new quantity, as indicated by Fig.2. Nevertheless, there seems to be some sufficiently ‘simple’ underlying mathematics, as displayed in the table. On the other hand, I am a bit skeptical concerning the claimed ‘universality’ of the results. This strong claim is clearly stated in the title already. I appreciate that the authors provide some evidence for universal features by studying a variety of symmetric random walks exhibiting different properties (recurrence, marginality, transience) in different dimensions. But the analytical approaches yielding the results in the table are certainly not ‘rigorous’ but rely on many more or less controlled approximations (that the authors try to justify in their ms. as much as possible). The numerical results, in turn, were only obtained for a rather selected collection of different random walk models, where I am not sure according to which criteria precisely they have been chosen (see below for more details). It is thus not clear to me on which assumptions the presented results really hold, apart from the processes being Markovian, which already sorts out quite a large number of other prominent random walk models.

Apart from this main concern, I have a number of other, more detailed remarks that I would like to convey to the authors, as included below. In summary,

I emphasize again that this is a very interesting article that many readers may find inspiring. But as it stands, I am quite on the fence here regarding its importance and impact for publication in Nature Commun. In my view, much more (sound) evidence needs to be provided for the claimed universality of the results.

1. abstract: To me the term ‘distinct site’ was not so clear, as it was only defined with respect to two sites and could have included returns. It only became clear to me in the first column when it was clarified that indeed ‘new’ sites were meant. Perhaps that should be explained right away.

2. p.1, right column, bottom: I may remark that the term ‘aging’ used here for the exploration time distribution is at least methodologically a bit different from what has been discussed on other occasions within stochastic theory, see, e.g., Metzler et al., Phys. Chem. Chem. Phys. 16, 24128 (2014). Here ageing is defined as the ‘dependence of physical observables [...] on the time span between initialisation of the system and the start of the measurement.’ Perhaps that can be clarified a bit in this ms. by giving a reference.

3. Fig.2: I found this figure partially a bit confusing. First one might think that subfigure (a) is a blowup of (b), but it is not. The meaning of the red circles is actually not explained. Then, the region shown in subfigure (a) is clearly bounded, nevertheless it is said in the caption that the visited region is ‘effectively infinite’ (see also further below). This looks a bit odd. The color code partially seems to relate to the table but not completely.

4. p.2, left column: Here it seems some conditions are given under which the claimed universality is supposed to hold. But from the analytical arguments it seems clear that non-Markovian dynamics is excluded, which in turn excludes prominent examples of random walks like Lévy walks and subdiffusive continuous time random walks (though there is some debate in the literature whether CTRWs are Markovian or not). Apart from symmetry and Markovianity, any further conditions that the authors would assume to be necessary for having Table 1? I think it would be important to make these clear.

5. p.3, right column: I am a little bit puzzled again by the assumption that for sufficiently short times the visited domain must be ‘essentially infinite’. If I understand correctly, the idea is that this way one has the situation of a large visited region that contains ‘traps’ as holes, see Fig.2(a), so that one can apply a combination of first passage time argument to find a trap region, and the return to the interface from within a trap region, see Eq.(1). But for this to hold it seems a sufficiently long time must have passed already, from the point where the particle just started to explore without having visited many points at all, which in turn seems to define another short time regime that precedes the scenario in Fig.2(a)? I guess there is nothing to say about this very short time dynamical regime then, at least not in terms of ‘universal’ features, as it may be very different for different types of random walks (say, Lévy flights compared to nearest neighbour random walks, for example)?

6. Eq.(1): Perhaps P_{trap} needs to be explained right after the equation.

7. p.4, right column: I find the explanation of the stretched exponential decay a bit short.

8. p.5: Why are only superdiffusive Lévy flights in one dimension considered, i.e., not in higher dimension (there are various different models for them), and not ballistic ones? I am actually surprised that even for them Table 1 holds, as due to the long jumps (no nearest neighbour process) there should be a lot of 'holes' (or traps) in the patterns of visited sites. And what about Lévy walks? Altogether I am not quite sure about the collection of random walks selected here to demonstrate universality. While I appreciate that they represent different properties, to some extent, if I compare this selection with what one of the authors did in Nat. Phys. (2015) for cover times of random searches (persistent, intermittent, Lévy walk in 1, 2, 3 dimensions), the present arsenal seems way less generic. I really do think more evidence has to be given for supporting supposedly universal features, especially as the analytical arguments (again, to be appreciated, and very skillful) are not rigorous. I think if this could be accomplished, it would make the paper much stronger.

9. The growth of the perimeter in Fig.5 reminds me of theoretical work on what is called the convex hull of a random process by Majumdar and others (while if I remember correctly, this can only grow while the perimeter can shrink, nevertheless there may be some asymptotic relation).

10. Has something like the exploration time distribution already been measured in experiments, or can the authors conceive experiments where this should be done? Foraging and self-interacting random walks are mentioned in the introduction, but details remain unclear.

11. Related to the previous remark there remains of course the problem of how this concept is translated to a process that is continuous in time and space, and not defined on a lattice. Any ideas?

I. REFEREE 1

We thank the referee for their careful reading of the manuscript and constructive comments. We reproduce this report below in red and our responses are interleaved.

In the present manuscript the authors explore in a very general form the dynamics of the time between visits to newly visited sites for a large class of random walk processes. They derive different regimes for recurrent, marginal and transient walks, and provide corresponding scalings of the intervisit time distributions. They do this by mapping this problem into a trapping problem with distributed traps and introducing several assumptions in order to deal with the temporal and spatial correlations present in the shape of the walker's visited/empty regions. Also, numerical results are presented for a variety of random walk models/simulations in order to support the validity of the different regimes reported analytically.

In my opinion, the whole work seems to be technically correct and sound, and it is written in a clear and precise way. My main concerns are about the real relevance and implications of the results presented. In particular, I do not feel able to assess properly that relevance until the authors have clarified two main points:

(i) The numerical results reported confirm that the scalings found for the different regimes provide a good fit to a large class of random walks. However, in my opinion this does not necessarily mean that all the previous assumptions made to obtain such results are completely valid. It could be that such scaling emerge somehow even if those hypothesis are only partially valid (or even wrong). So, in my opinion the authors should provide a more exhaustive numerical analysis checking directly (not indirectly) the validity of those assumptions.

In our previous manuscript version, some assumptions were numerically checked only indirectly. In this revision, we provide now an exhaustive numerical analysis to check the validity of the following points (as indicated explicitly in the updated manuscript):

- **Distribution $Q_n(r)$ of the radius of the largest spherical region free of traps after n sites have been visited:** in section S2.D of the SI, we provide a direct check of (i) the scaling form of $Q_n(r)$, (ii) the behavior with n of the radius ρ_n of the largest region free of traps and (iii) the exponential tail of $Q_n(r)$, as given in the main text. These points are the key intermediate results underlying our final results on the distribution $F_n(\tau)$.
- **First regime appearing in recurrent and marginal random walks:** we provide a direct check of our claims that (i) the region free of traps is seen as infinite in this regime (see the independence and the data collapse with respect to the number of sites visited n in Fig. 3 of the main text and Supplementary Fig. 7a of the SI), (ii) the distribution F_n is algebraic (see Figs. 3a–e of the main text and Supplementary Fig. 7a in Section S4.A in the SI), (iii) the value of the corresponding exponent is $1 + \mu$ (see the red dashed line slope in Fig. 3a to e of the main text and Supplementary Fig. 7a in Section S4.A in the SI), (iv) the value of the corresponding prefactor A can be determined exactly in the marginal case and approximately in the recurrent case (see the dashed line in Fig. 3 of the main text and Supplementary Fig. 7a in Section S4.A in the SI), (v) the fractal dimension of the perimeter $P(t)$ of the visited domain is given by Eq. (S29) of the SI (see Section S3.A, Supplementary Fig. 5).
- **Scaling with n of the crossover time t_n up to which this first regime holds:** This behavior is confirmed by (i) the scalings with n of both the moments $\langle \tau_n^k \rangle$ and the rate θ_n of the exponential decay at large times (see Section S3.D Supplementary Fig. 6a, b and c, and Section S4.A Supplementary Fig. 7 of the SI), (ii) Fig. 3a, b and c of the main text showing that for recurrent random walks $t_n = T_n$ as the rescaling of time τ/θ_n makes all the curves collapse, (iii) Fig. 3e and f of the main text, which shows that for marginal random walk $t_n = \sqrt{n}$ as the rescaling of time τ/\sqrt{n} makes all the curves collapse, (iv) for the marginal case, the fact that the first moment of τ_n for large n given by $\frac{1}{\pi} \ln n = \frac{2}{\pi} \ln t_n$, and higher moments by $\langle \tau_n^k \rangle \sim n^{(k-1)/2}$ as shown in Section S3.D Supplementary Fig. 6c and d of the SI, (v) the absence of algebraic regime for transient walks, as shown in Fig. 3g and h in the main text and the convergence at large n of the moments shown in Section S3.D Supplementary Fig. 6e and f.
- **Second regime appearing in marginal and transient random walks:** we directly check our claims that the distribution F_n is stretched exponential with the exponent given in Table I of the main text (as the log probability divided by $(\tau/t_n)^{\mu/(1+\mu)}$ plotted in Fig. 3f to i of the main text converges to a constant value). This constitutes a check of our claim that the asymptotic limit of the lower bound (3) given in the main text indeed provides the right intermediate time behaviour. We added in SI Secs. S4.C the examples of transient Lévy flights, which further confirms this feature.

- **Scaling with n of the crossover time T_n up to which this second regime holds:** It is confirmed by (i) the data-collapse of both the marginal and transient walks using the rescaling τ/T_n in Fig. 3f and i of the main text and Supplementary Fig. 9 in Section S4.C of the SI, (ii) the collapse to the exponential regime for RWs on hypercubic lattices at $\tau \sim T_n$ (see Fig. 3f and i of the main text, as well as Supplementary Fig. 9 in Section S4.C of the SI).
- **Covariance of the number of distinct sites visited:** We verify the proposed scalings in times and check the hypothesis that correlations between τ_n can be neglected in SI Section 5.B. In the case for which the hypothesis is not verified, we prove the existence of a lower bound for any recurrent random walk. Details on these aspects are provided on the next page.

To sum up, we followed the Referee recommendation and checked all assumptions that underlie our results.

I am particularly intrigued about the assumption made in the derivations in Section S2 that "behavior of all nodes in the ball of radius r^γ is equivalent, and the exploration of these nodes are disjoint events". I do not find this intuitive at all, so I would urge the authors to provide evidences that such assumption is really sound.

We realize that this point was not checked directly. We provide here (see also SI Sec. S2): (i) a derivation of the behavior of the typical number $k_c(r)$ of incursions needed to visit all sites within a ball of radius r^γ/e^2 in the marginal case, as well as (ii) a direct numerical check of this behavior. Note that the unimportant constant factor e^2 appears only for convenience, to ease comparison with numerical simulations, and does not modify the behavior of $k_c(r)$ for large r , as is clear from the proof below.

We first recall the definition of a γ -incursion, an incursion inside a ball of radius r^γ that hits a ball of radius $a = r^\gamma/e$ before a ball of radius $b = r$. The incursion ends when the RW returns to the ball of radius r^γ . We note that the probability of not reaching the origin during any of k γ -incursions is given by

$$\left(1 - \frac{1}{\gamma \ln r}\right)^k \sim \exp\left(-\frac{k}{\gamma \ln r}\right).$$

Because there are $(r^\gamma/e^2)^{d_f}$ sites inside a ball of radius r^γ/e^2 , the probability of having at least one non-visited site after k incursions can be estimated as

$$\begin{aligned} & 1 - \mathbb{P}(\text{all sites are visited in the ball of radius } (r^\gamma/e^2)^{d_f}) \\ & \approx 1 - \mathbb{P}(0 \text{ is visited})^{(r^\gamma/e^2)^{d_f}} \\ & = 1 - \left(1 - \exp\left(-\frac{k}{\gamma \ln r}\right)\right)^{(r^\gamma/e^2)^{d_f}} \\ & \approx (r^\gamma/e^2)^{d_f} \exp\left(-\frac{k}{\gamma \ln r}\right) = \frac{1}{e^{2d_f}} \exp\left(\gamma d_f \ln r - \frac{k}{\gamma \ln r}\right) \end{aligned} \quad (1)$$

by assuming that all $(r^\gamma/e^2)^{d_f}$ sites have the same probability to be visited as the origin, and that the visits to these sites are *independent* events. There was indeed a misprint in the previous manuscript version, where it was written that these events were "disjoint". We thank the Referee for spotting this misprint.

This calculation results in the value $k_c(r) = d_f \gamma^2 (\ln r)^2$. We numerically check this result for $k_c(r)$ for marginal $1d$ Lévy flights in Fig. 1 where we plot versus $\ln r$ the average number $k_c(r)$ of γ -incursions (with $\gamma = 1$) inside the ball of radius r/e starting at the position $\pm r/e$ (with equal probability) to have visited all of the points in the ball of radius r/e^2 . We note that the convergence to the asymptotic $(\ln r)^2$ dependence occurs only for large r .

Additionally, as explained in the previous version of the SI, in the case of (marginal) $2d$ regular random walks, the above result is supported by a theorem in the mathematical literature [1]. In our revision, we provide simple physical arguments that allow to extend the above scaling behavior for $k_c(r)$ to any marginal random walk.

To sum up, in the new version of the SI, we corrected our misprint about the "disjoint" events hypothesis, clarified its origin as well as the analytical result concerning $k_c(r)$ and provided a direct numerical check of the scaling of $k_c(r)$ with r .

The same would apply to the assumption of independence between 'tau' variables in Section S5.B, which is only checked indirectly.

FIG. 1. $k_c(r)$ compared to $(\ln r)^2$ (black dashed line).

FIG. 2. Summed covariance divided by $n_1^{2\alpha}$, $Y \equiv \frac{1}{n_1^{2\alpha}} \text{Cov} \left[\sum_{k=0}^{n_1-1} \tau_k, \sum_{k=n_1}^{n_2-1} \tau_k \right]$ as a function of n_2/n_1 for two different values of $n_1 = 10$ (blue stars) and $n_1 = 100$ (orange circles) for Lévy flights of parameter **a** $\alpha = 1.3$, **b** $\alpha = 1.5$, **c** $\alpha = 1.7$.

We acknowledge that this point needed clarification. As we show below (see also Section S5.B of the SI), the result on the covariance Eq. (6) of the number $N(t)$ of distinct sites visited by a recurrent random walk given in the main text does not require strict independence of the variables τ_n , but only a weaker hypothesis on their correlations. We provide in the new version of the SI: (i) this quantitative criterion (see Eq. (2) below) and (ii) a direct numerical check that this criterion is indeed satisfied for $1d$ recurrent Lévy flights (see Supplementary Fig. 13, section S5.B of the SI).

The main result can be summarized as follows. Starting from the relation between $N(t)$ and the sum of τ_k , $\{N(t) \geq n\} = \left\{ \sum_{k=0}^{n-1} \tau_k \leq t \right\}$, we perform a cumulant expansion which gives the following decorrelation criterion,

$$\text{Cov} \left[\sum_{k=0}^{n_1-1} \tau_k, \sum_{k=n_1}^{n_2-1} \tau_k \right] = O \left(n_1^{2/\mu} \right), \quad (2)$$

for large values of the number of distinct sites visited n_1 and $n_2 > n_1$. If this criterion is respected, one gets Eq. (9) of the main text.

Fig. 2 provides a direct numerical check that this criterion is satisfied by recurrent $1d$ Lévy flights, for a range of values of $1 < \alpha < 2$ (for $1d$ Lévy flights, $\mu = 1/\alpha$). The physical interpretation of the criterion of Eq. (2) is that, even if the number of holes in the support of the random walk diverges with time, their relative number compared to the number of distinct sites visited is small. We thus effectively have a $1d$ problem with asymptotically no holes, for which the variables τ_n are independent. The criterion in Eq. (2) makes this qualitative argument quantitative.

Beyond this case of $1d$ Lévy flights, where the correlations can in practice be neglected, the new calculation performed in SI Section S5.B allows us to obtain the following general lower bound for the covariance of $N(t)$ of recurrent random walks

$$\text{Cov} [N(t_1), N(t_2)] \geq t_1^\mu t_2^\mu \frac{t_1}{t_2}. \quad (3)$$

As explained in the revised manuscript, this bound allows us to show that temporal correlations of the number of distinct sites visited by a recurrent random walk have a long memory (see also the next point).

(ii) Regarding the relevance and impact that studying the distribution of elapsed times between visits may have, the arguments provided in the introduction are not very convincing. While it is true that times between new visits are more fundamental magnitudes than the mean number of visited sites or other similar measures of search exploration, it is not clear whether this line of research has the potential to generate a shift in the interest of workers in the field. In my opinion, the arguments provided in the last part of the main text (about the opportunities to explore multiple-time correlations between visits) are promising at this respect, but they are only barely sketched in the manuscript. If the authors really believe that this is a real possibility for future research, they should provide more tangible arguments than those simply based on the correlations of $N(t)$ for Lévy flights (again, I have my concerns about the assumptions made in Section S5.B) to support it.

We acknowledge that the arguments provided in the manuscript concerning the relevance and impact of the statistics of times between visits needed improvement. In response to the Referee’s objection about the relevance and impact of our study, we extended our manuscript by discussing the two important examples of self-attracting RWs and of the foraging theory. We address these issues in points (ii)-(iii) of our response to Referee 2 on page 6. Finally, to provide more tangible arguments that our research has the potential to spark the interest of researchers in the field, we significantly expanded the section about the applications of our results to determining temporal correlations in the number of distinct sites visited. See point (i) in the response to Referee 2 for details.

To sum up, we now show that our results are useful to reanalyse the question of the number of distinct sites visited, and our results have applications to both self-interacting RWs and foraging theory. To return to the question of future research raised by the Referee and also by Referee 3, we also investigated non-Markovian random walks, such as fractional Brownian motion (both sub- and super-diffusive), and the true self-avoiding walk in our revision. Our formalism can deal with these complex situations. In particular, the exponent characterizing the algebraic decay in recurrent cases still applies to these non-Markovian examples, up to the replacement $\mu \rightarrow 1 - \theta$ where θ is the persistence exponent (see Section S5.C of the SI). This intriguing fact adds to the potential for future research opened by our work.

Minor point:

iii) The authors provide a short discussion about the dispersion of τ values in page 2 of the main text, but very few is shown about this. For the sake of completeness this is an issue that should be explicitly shown, at least as a part of the Supplementary Material.

We agree that results concerning the moments of the τ_n were missing in the original manuscript. In the new section of the SI S3.D, we now provide a comparison between the analytical results and numerical simulations of these moments.

II. REFEREE 2

The manuscript by Regnier et al studies the statistics of time intervals between visits of two consequent previously unvisited sites by a random walk. Results of the analytical evaluation (mostly asymptotic analysis of various regimes) are validated with numerical simulations. A broad range of examples of random walks from 1D Levy flights to fractal structures on 2D lattice and hypercubic lattices, is considered. The paper is clearly written, with only most relevant key results retained in the main text of the manuscript while all derivations and technical steps are delegated to the supplemental material.

In the context of this manuscript being submitted to Nature Communications I have the following critical comments. In essence, the manuscript is a meticulous analysis of different regimes of the interval statistics for different types of random walks (as defined by the parameter μ). I totally enjoyed reading it as a person working in this field and learned new things. While I also totally agree with the fundamental mathematical nature of the problem studied, the way it is presented does not convince that this problem is of broad interest as implied by the Nature Communications standards. The manuscript is more of a format of a specialised journal, like PRE or Journal of Statistical Physics. The authors claim that the quantity analysed is “fundamental and remarkably useful” but why is it so is never actually shown and discussed. I also actually doubt that one could claim anything like that objectively. Motivation for the analysis of this quantity is also not convincing, which is similar to the argument that the statistics of the distinct sites visited by a random walk is important and thus also the time intervals between the two most recent visited new sites. No practical examples of the importance and/or relevance of the presented results were provided (except for the rather

vague discussion on cells with metabolic clocks). Here, the fact that RWs are ubiquitous across disciplines, does not work as a strong argument for the broad interest in the particular quantity studied in this work. To summarise, I do not think that this work is of general importance and interest to the broad readership of Nature Communication and therefore can not recommend it for publication.

We thank the Referee for their careful reading of the manuscript and their constructive comments. We acknowledge that the arguments we originally provided concerning the relevance and impact of the statistics of times between visits needed improvement. In this revision, to respond the Referee's concern we make the following three points:

- (i) We significantly expanded the section devoted to the applications of our results to the temporal correlations of the number of distinct sites visited. We now provide:
 - A quantitative criterion to neglect the correlations between the τ_n in determining the covariance of $\{N(t)\}$ as well as a direct numerical check (see also the response to the point (ii) raised by Referee 1).
 - An extension of our approach to determine not only the covariance, but any multiple-time correlation functions of $\{N(t)\}$. We consider this to be an important result as it allows us to quantify the full stochastic process $\{N(t)\}$ through its n -time moments.
 - A new figure in the main text that displays the agreement between our theoretical prediction and numerical simulations both for the 2- and 4-time correlations in the case of recurrent Lévy flights (for which the above quantitative criterion is verified).
 - An extension of our results on the n -time correlations to general recurrent random walks, by giving a general lower bound that holds even when the correlations between the τ_n cannot be neglected. This finally allows us to show that temporal correlations of the number of distinct sites visited by a recurrent random walk always have a long time memory.
- (ii) We better motivate the importance of the times between visits to new sites. In the introduction, we now show how the variables τ_n are relevant to self-interacting random walks, by giving details on the so-called self-attracting random walk. Here, a random walker deposits a signal at each visited site, which alters the future dynamics of the walker on its next visits. This self-attracting walk has recently been shown to account for real trajectories of living cells, and is thus of particular interest.

The precise link with the times between visits of new sites is as follows. In this self-attracting walk, the probability that the RW jumps to a neighboring site i is proportional to $\exp(-u n_i)$, where $n_i = 0$ if the site i has never been visited up to time t and $n_i = 1$ otherwise. The analysis of this non-Markovian walk is notoriously difficult, with few results in dimension higher than 1. However, we stress that the random walk evolution between the visits of new sites, of durations *precisely* given by the variables τ_n , is described by a *regular* random walk whose properties are well known. The evolution of the walk between visits to new sites thus is simple, and the τ_n are the key variables to analyse this problem. Even if obtaining the full statistics of the position of such a RW also requires knowledge of the correlations between times between visits of new sites and the position of the new sites visited, knowledge of the statistics of the τ_n represents an important first step in analysing and understanding these non-Markovian RWs.

- (iii) As an application to foraging theory, we now answer the question raised in the introduction about the lifetime of a $2d$ starving random walker. Such a walk survives only if the time elapsed until a new food-containing site is visited is smaller than an intrinsic metabolic time \mathcal{S} . The forager collects a unit of resource each time a new site is visited. In a given trajectory, the forager might find resources at an almost regular rate, while in another trajectory the forager might find most of its resources near the end of its wandering. This discrepancy in histories has dramatic effects: if the time until a new food-containing site is reached is smaller than \mathcal{S} , the forager survives on the first trajectory but not on the latter. While the forager lifetime is known in $1d$ and in mean-field models, the answer in $2d$, relevant to most applications in foraging theory, was open. Based on our results for the τ_n , we answer this question in our revision: the mean lifetime of a starving RW grows *quadratically* with the metabolic time \mathcal{S} (much faster than the linear growth with \mathcal{S} in $1d$). We checked this analytical result numerically. A paragraph and a figure have been added to the revised manuscript to highlight these aspects.

To sum up, we now clearly show that our results are useful to reanalyse the old question of the number of distinct sites visited and have applications both for self-interacting RWs and foraging theory. Thus we argue that our work is of general importance and interest to the broad readership of Nature Communications.

Finally, driven also by a question from Referee 3, we studied, in our revision, examples of non-Markovian random walks, including strongly non-Markovian ones (like the fractional Brownian motions and the True Self-Avoiding

Walk). Interestingly, our results still describe these complex situations. In particular, the exponent characterizing the algebraic decay in recurrent cases still applies to these strongly non-Markovian examples. We believe that this intriguing fact adds to the potential for future research opened by our work.

III. REFEREE 3

The authors introduce and explore an, as far as I can tell, novel quantity to characterise the spreading dynamics of random walks, which is the probability distribution of the elapsed time between visits of a walker to distinct (new) sites. It is intuitively evident that this quantity is highly non-trivial and non-stationary, as it characterises the stochastic process in terms of the ‘pattern’ of different sites visited in time, which in turn changes profoundly with time. They calculate this distribution for symmetric random walks by a combination of analytical methods, as explained in detail in the Supplemental Material. These approximate analytical results are supported by computer simulations for a variety of selected random walks models. The key results are summarised in table 1, which gives the asymptotic time dependence of the ‘exploration (or visitation) time distribution’ (as one may call it) for different regimes of time, and for different fundamental classes of random walks. It is claimed that these results are universally valid for symmetric random walks. I find the article very well written. The main ideas are clearly summarised in the main text while all technical details are shifted into the Supplement, which indeed shows a lot of knowledge and skills. Figure 1 nicely represents the main idea of this approach, Fig.2 together with the table yields a concise summary of the main results including the underlying physics. The essence of this work is thus well accessible to a general physics readership. I like the main idea underlying this research, which is to study the ‘exploration time distribution’. This quantity indeed goes beyond more conventional approaches characterising random walks by position distributions, associated mean square displacements, or first passage time distributions. Obviously there is a lot of physical content in this new quantity, as indicated by Fig.2. Nevertheless, there seems to be some sufficiently ‘simple’ underlying mathematics, as displayed in the table. On the other hand, I am a bit skeptical concerning the claimed ‘universality’ of the results. This strong claim is clearly stated in the title already. I appreciate that the authors provide some evidence for universal features by studying a variety of symmetric random walks exhibiting different properties (recurrence, marginality, transience) in different dimensions. But the analytical approaches yielding the results in the table are certainly not ‘rigorous’ but rely on many more or less controlled approximations (that the authors try to justify in their ms. as much as possible). The numerical results, in turn, were only obtained for a rather selected collection of different random walk models, where I am not sure according to which criteria precisely they have been chosen (see below for more details). It is thus not clear to me on which assumptions the presented results really hold, apart from the processes being Markovian, which already sorts out quite a large number of other prominent random walk models. Apart from this main concern, I have a number of other, more detailed re-marks that I would like to convey to the authors, as included below. In summary, I emphasize again that this is a very interesting article that many readers may find inspiring. But as it stands, I am quite on the fence here regarding its importance and impact for publication in Nature Commun. In my view, much more (sound) evidence needs to be provided for the claimed universality of the results.

We thank the Referee for their careful reading of the manuscript and their constructive comments. Their main recommendation to improve the manuscript is as follows: “I really do think more evidence has to be given for supporting supposedly universal features, especially as the analytical arguments (again, to be appreciated, and very skillful) are not rigorous. I think if this could be accomplished, it would make the paper much stronger”.

We have taken this recommendation seriously, and modified our manuscript accordingly. To support the universal features of our results:

- We provide further evidence for our claims, for additional examples of Markovian random walks (see Fig. 4 of the main text and SI Sec. S4.C): (i) transient Lévy flights in $1d$ (in the previous version of the manuscript, only recurrent ones were considered), (ii) $2d$ and $3d$ Lévy flights and (iii) persistent random walks, both in $2d$ and $3d$.
- We also extended the initial framework of our approach to cover the prominent examples mentioned by the Referee of (i) CTRWs (see Supplementary Fig. 15 in SI) and (ii) $1d$ Lévy walks (see Supplementary Fig. 17 of the SI).
- Driven by the Referee’s challenging questions, we investigated non-Markovian models (in the recurrent case), which we added in the main text, such as the $1d$ fractional Brownian motion (fBm) and the $1d$ true self-avoiding walk (TSAW), which have infinite memory. Amazingly, these non-trivial examples are still described by our results. We added an illustrative figure in the main text (see Fig. 4) on this point.

We believe that these additions to the manuscript unambiguously address the criticism raised by the Referee about lack of evidence for the universality of our results.

We now provide a point-by-point response to the additional comments raised by the Referee.

1. abstract: To me the term ‘distinct site’ was not so clear, as it was only defined with respect to two sites and could have included returns. It only became clear to me in the first column when it was clarified that indeed ‘new’ sites were meant. Perhaps that should be explained right away.

To make the definition clearer, we modified the abstract to emphasize that τ_n is the elapsed time to find *new* site when n sites have already been found.

2. p.1, right column, bottom: I may remark that the term ‘aging’ used here for the exploration time distribution is at least methodologically a bit different from what has been discussed on other occasions within stochastic theory, see, e.g., Metzler et al., Phys. Chem. Chem. Phys. 16, 24128 (2014). Here ageing is defined as the ‘dependence of physical observables [...] on the time span between initialisation of the system and the start of the measurement.’ Perhaps that can be clarified a bit in this ms. by giving a reference.

We added reference [2] (Ref. [44] in the main text), which contains the definition of aging we use. Indeed, according to this paper, “Aging media are generally characterized by the fact that physical observables depend on the time elapsed since the preparation of the system”. In our work, the time elapsed since the preparation of the system is the time required to have visited n sites. Once conditioned on having reached this time, we look at the properties of the system of n distinct sites visited, in particular the first exit time of the visited domain. As shown in the main text, this observable strongly depends on the choice of n and thus the time elapsed since the preparation of the system. This point has been clarified in the new version of the main text.

3. Fig.2: I found this figure partially a bit confusing. First one might think that subfigure (a) is a blowup of (b), but it is not. The meaning of the red circles is actually not explained. Then, the region shown in panel (a) is clearly bounded, nevertheless it is said in the caption that the visited region is ‘effectively infinite’ (see also further below). This looks a bit odd. The color code partially seems to relate to the table but not completely.

We modified the following in the caption of Fig. 2 and Table 1 to account for these criticisms: (i) Indeed, **a** is not a blowup of **b**, they are different controlling configurations of the different time regimes. (ii) The circles around each figure is meant to correspond to the time regimes identified in the table: **a** corresponds to the time regime $1 \ll \tau \ll t_n$ (in red), **b** to the time regime $t_n \ll \tau \ll T_n$ (in green) and **c** to the time regime $T_n \ll \tau$ (in blue). This color code is also used in Figs. 3 and 4. (iii) The region shown in **a** is bounded for practicality, but we stress that it is effectively infinite at the scale of the trajectory that the RW makes to visit the new site. In order to emphasize the unbounded aspect, we replaced the red circle around Fig. 2a by a circle with red dashed lines. (iv) We emphasize that the color code relates to the table, as we now stress in the caption of Table 1.

4. p.2, left column: Here it seems some conditions are given under which the claimed universality is supposed to hold. But from the analytical arguments it seems clear that non-Markovian dynamics is excluded, which in turn excludes prominent examples of random walks like Lévy walks and subdiffusive continuous time random walks (though there is some debate in the literature whether CTRWs are Markovian or not). Apart from symmetry and Markovianity, any further conditions that the authors would assume to be necessary for having Table 1? I think it would be important to make these clear.

Indeed, we use the Markovian assumption to obtain the results summarized in Table 1. Apart from symmetry and the Markov nature, we rely on discrete time and space RWs, and consider processes for which the walk dimension d_w is well defined (which is not a very restrictive assumption). This is now explicitly stated in the main text. However, in this revised version, we present extensions of our results to continuous-time RWs and to some non-Markovian processes (see below).

5. p.3, right column: I am a little bit puzzled again by the assumption that for sufficiently short times the visited domain must be ‘essentially infinite’. If I understand correctly, the idea is that this way one has the situation of a large visited region that contains ‘traps’ as holes, see Fig.2(a), so that one can apply a combination of first passage time argument to find a trap region, and the return to the interface from within a trap region, see Eq.(1). But for this to hold it seems a sufficiently long time must have passed already, from the point where the particle just started to explore without having visited many points at all, which in turn seems to define another short time regime that precedes the scenario in Fig.2(a)? I guess there is nothing to say about this very short time dynamical regime then, at least not in terms of ‘universal’ features, as it maybe very different for different types of random walks (say, Lévy

flights compared to nearest neighbour random walks, for example)?

Indeed, the short-time behavior can be distinct for random walks that share the same μ parameter; for example, the $2d$ simple random walk and the $1d$ Lévy flight with parameter $\alpha = 1$ (which both correspond to $\mu = 1$). For the first case, there exist trajectories where the RW is sure not to find a new site (for example if it is surrounded by already visited sites). This situation never happens for the second case because the walk can escape such a region by a long jump. These features affect only the short-time dynamics ($\tau = O(1)$) and depend on the details of the models. This is why we only deal with times τ_n long enough so that universal features arise (see footnote [52] of the main text).

6. Eq.(1): Perhaps P_{trap} needs to be explained right after the equation.

We added a comment before equation (1) to clarify this point.

7. p.4, right column: I find the explanation of the stretched exponential decay a bit short.

We added details in the main text concerning the explanation of the stretched exponential decay.

8. p.5: Why are only superdiffusive Lévy flights in one dimension considered, i.e., not in higher dimension (there are various different models for them), and not ballistic ones? I am actually surprised that even for them Table 1 holds, as due to the long jumps (no nearest neighbour process) there should be a lot of ‘holes’ (or traps) in the patterns of visited sites. And what about Lévy walks? Altogether I am not quite sure about the collection of random walks selected here to demonstrate universality. While I appreciate that they represent different properties, to some extent, if I compare this selection with what one of the authors did in Nat. Phys. (2015) for cover times of random searches (persistent, intermittent, Lévy walk in 1, 2, 3 dimensions), the present arsenal seems way less generic. I really do think more evidence has to be given for supporting supposedly universal features, especially as the analytical arguments (again, to be appreciated, and very skillful) are not rigorous. I think if this could be accomplished, it would make the paper much stronger.

We have taken this recommendation seriously. As written at the beginning of the response to the referee, we consider more generic examples of random walks. All of them confirm our results. We added the corresponding figures in the main text and the SI. We believe that the new version of the manuscript unambiguously addresses the main criticism raised by the Referee of lack of evidence of the universality of our results.

Concerning the comparison with the article on cover times, we stress that the examples covered there (Nat. Phys. (2015)) were all *transient*. The range of applications, and the examples we treat in this revision, is much broader.

9. The growth of the perimeter in Fig.5 reminds me of theoretical work on what is called the convex hull of a random process by Majumdar and others (while if I remember correctly, this can only grow while the perimeter can shrink, nevertheless there may be some asymptotic relation).

In our work, we deal with the total perimeter of the visited domain—both the interior (perimeter of islands of non visited sites inside the domain) and external frontier of the visited domain. The convex hull only deals with the later, which is much shorter than the total perimeter. In [3] it was shown that the perimeter of the convex hull after t steps of the random walk is \sqrt{t} while the total perimeter we consider here is [4] $t/(\ln t)^2$. Even though both observables are useful to quantify the territory explored by a random walker, they are distinct. Studying the growth of the perimeter of the convex hull could be an interesting question, which we leave for future work.

10. Has something like the exploration time distribution already been measured in experiments, or can the authors conceive experiments where this should be done? Foraging and self-interacting random walks are mentioned in the introduction, but details remain unclear.

We are not aware of such experiments. One might test foraging by taking a bacteria, placing it in a tube containing food everywhere, look at its starvation time, and compare it to the situation where the bacteria is on a petri dish. Bacteria should have no chemotaxis mechanism, and death time would be dominated by starvation and not external conditions. The example of self-attracting random walks is now made fully explicit in the introduction. The application to foraging is now considered in detail at the end of the main text of the revised version.

11. Related to the previous remark there remains of course the problem of how this concept is translated to a process that is continuous in time and space, and not defined on a lattice. Any ideas ?

The generalisation of the number of distinct sites visited to continuous time and space is known as the Wiener

sausage [5]. Many results for visitation in discrete random walks carry over to those obtained for the Wiener sausage (see [6]). In this setting, τ_n is the elapsed time between the covering (by the Wiener sausage) of a domain of volume n and volume $n + 1$. This definition naturally induces a discretization of the time. Because of the close correspondence between the continuous and discrete problems and because the continuous problem raises no open questions, we don't think it is worth pursuing in our current manuscript.

-
- [1] A. Dembo, Y. Peres, and J. Rosen, *Ann. Probab.* **35** (2007).
 - [2] N. Levernier, O. Bénichou, T. Guérin, and R. Voituriez, *Phys. Rev. E* **98**, 022125 (2018).
 - [3] G. Claussen, A. K. Hartmann, and S. N. Majumdar, *Phys. Rev. E* **91**, 052104 (2015).
 - [4] F. van Wijland, S. Caser, and H. J. Hilhorst, *J. Phys. A: Math. Gen.* **30**, 507 (1997).
 - [5] M. D. Donsker and S. S. Varadhan, *Commun. Pure Appl. Math.* **28**, 525 (1975).
 - [6] A. Berezukovskii, Y. A. Makhovskii, and R. Suris, *J. Stat. Phys.* **57**, 333 (1989).

REVIEWERS' COMMENTS

Reviewer #1 (Remarks to the Author):

After reading the new version of the manuscript, I have to acknowledge that the authors have carried out a considerable effort in order to satisfy my (as well as the other referees') requirements. The new version then clarifies some aspects that were probably only partially sketched in the original one.

Going to the specific points I mentioned in my first report, I must say that the second point in there (about the potential interest/applicability of the results to the researchers in the field) has been addressed in a rather satisfactory way. The authors now do not merely restrict their discussion to the Lévy case in 1d, but they use scaling properties of recurrent random walks plus a cumulant expansion to provide a rather general criteria (reported in the new equation 8) to discern when the covariances in the number of sites visited up to some time are expected to follow expression 6. In my opinion this is a quite valuable result/illustration of how the present work can be used as a reference for researchers in the field to address the problem of time correlations during the coverage process.

The dispersion/variability of tau's now is studied in a more thorough way. Although much more could be ideally said about this, I understand that space limitations and readability constraints make that the analysis provided is now enough as an introduction to this point.

I am much less satisfied, however, about the response provided to the other main point in my report. I clearly stated that my concern was about the validity of the fundamental hypotheses used to derive the scaling of the tau_n pdf's. In particular, the hypothesis about (a) 'equivalence' of sites and (b) 'independent' (not disjoint, as recognised by the authors) visits to sites within a ball of radius r^{γ}/e^2 , are still introduced in the derivation ad hoc. The authors now provide a Figure in the SI file (Figure 2 of the SI) showing that new scaling properties predicted from those hypothesis approximately hold. Again, I argue that this is just an indirect proof. Besides, the scaling presented in Figure 2 of the SI is clearly not perfect, which confirms that the starting hypothesis are only partially valid in the best case. My concern is clear: are these scalings really a consequence of hypothesis (a) and (b), or do they emerge even though such hypothesis are not valid? In other words, how large the spatial and/or temporal correlations within the ball must be in order to break the scalings found?

Unless this last point is properly clarified I do not think that the robustness of the work and the results are fully proved, so I cannot provide my positive recommendation. Still, as I acknowledge that the authors have made a considerable effort in order to improve the manuscript in many other aspects, I would offer them a new opportunity to review this point (in case the editor accepts going through a third round of referral).

Reviewer #2 (Remarks to the Author):

In general, the authors have significantly advanced their manuscript in this revision. They also somewhat tried to address my criticism about the generality and significance of the new considered characteristics of random walks. While I mostly remain with my original assessment on that particular point, I don't feel that my disagreement should stand on the path of publication of this very professional work. Thus, I would be in favour of acceptance for publication.

Reviewer #3 (Remarks to the Author):

Title: Universal exploration dynamics of random walks

Authors: Léo Régnier, Maxim Dolgushev, S. Redner, and Olivier Bénichou

I thank the authors for their detailed, constructive reply to my first report. They have addressed all my minor comments satisfactorily. Concerning my main criticism, I asked for more evidence to support their claim about universality of their results. The authors did so by studying further examples of Markovian random walks, by including the prominent examples of CTRWs and Lévy walks that I mentioned, and by elaborating on generic, truly non-Markovian models, such as fractional Brownian motion. This led to adding an essentially new Fig.4 to the main text, while Fig.3 was substantially revised and updated accordingly. Furthermore, about 8 new pages have been added to the Supplement explaining the analysis of the newly added models, see Sec. S5 B and C.

I thus find that the authors have also suitably addressed my main query about universality by studying all these additional processes, and by delivering results that match to their theoretical framework as outlined before. Also in view of the comments raised by especially the first referee, I think the paper got substantially stronger. Consequently, I am now happy to recommend the present ms. for publication in Nature Communications as it is.

Response to Referee 1

We thank the referee for their careful reading of the manuscript and constructive comments. We reproduce this report below in red and our responses are interleaved.

After reading the new version of the manuscript, I have to acknowledge that the authors have carried out a considerable effort in order to satisfy my (as well as the other referees') requirements. The new version then clarifies some aspects that were probably only partially sketched in the original one.

Going to the specific points I mentioned in my first report, I must say that the second point in there (about the potential interest/applicability of the results to the researchers in the field) has been addressed in a rather satisfactory way. The authors now do not merely restrict their discussion to the Lévy case in 1d, but they use scaling properties of recurrent random walks plus a cumulant expansion to provide a rather general criteria (reported in the new equation 8) to discern when the covariances in the number of sites visited up to some time are expected to follow expression 6. In my opinion this is a quite valuable result/illustration of how the present work can be used as a reference for researchers in the field to address the problem of time correlations during the coverage process.

The dispersion/variability of tau's now is studied in a more thorough way. Although much more could be ideally said about this, I understand that space limitations and readability constraints make that the analysis provided is now enough as an introduction to this point.

We thank the Referee for the positive assessment of our revised manuscript.

I am much less satisfied, however, about the response provided to the other main point in my report. I clearly stated that my concern was about the validity of the fundamental hypotheses used to derive the scaling of the τ_n pdf's. In particular, the hypothesis about (a) 'equivalence' of sites and (b) 'independent' (not disjoint, as recognised by the authors) visits to sites within a ball of radius r^γ/e^2 , are still introduced in the derivation ad hoc. The authors now provide a Figure in the SI file (Figure 2 of the SI) showing that new scaling properties predicted from those hypothesis approximately hold. Again, I argue that this is just an indirect proof. Besides, the scaling presented in Figure 2 of the SI is clearly not perfect, which confirms that the starting hypothesis are only partially valid in the best case. My concern is clear: are these scalings really a consequence of hypothesis (a) and (b), or do they emerge even though such hypothesis are not valid? In other words, how large the spatial and/or temporal correlations within the ball must be in order to break the scalings found?

Unless this last point is properly clarified I do not think that the robustness of the work and the results are fully proved, so I cannot provide my positive recommendation. Still, as I acknowledge that the authors have made a considerable effort in order to improve the manuscript in many other aspects, I would offer them a new opportunity to review this point (in case the editor accepts going through a third round of referral).

We thank the Referee for offering us the opportunity to clarify this point.

To respond to the Referee's concern, we provide below (and in the revised version of Sec. S2.A.2 of the SI) a direct numerical check of the validity of hypotheses (a) and (b). As they concern the marginal case, we relied on numerical simulations of a 1d Lévy flight of parameter $\alpha = 1$.

We first confirm the hypothesis (a) of the equivalence of sites by directly measuring in numerical simulations the probability that a site x has been visited at the k^{th} γ -inursion inside a domain of radius r/e (taking $\gamma = 1$), starting at $\pm r/e$ and stopping the incursion when the ball hits the radius r . We denote this probability by $p_{x,r}(k)$. Then, hypothesis (a) is verified if $p_{x,r}(k)$ is independent of the site x . In Fig. 1, we take different values of x and r and verify that, indeed, at radius r large enough, the visitation statistics at the k^{th} incursion is independent of the site x .

We then confirm hypothesis (b) on the independence of sites by measuring in numerical simulations the correlation in the visitation events. More precisely, we look at the probability of having visited site 0 and site x before or at the k^{th} γ -inursion inside a ball of radius r/e ($\gamma = 1$). We denote this probability by $p_{0 \cap x, r}(k)$. In particular, we looked at the cases $k = 1$ and $k = 10$. Then, hypothesis (b) is verified if $p_{0 \cap x, r}(k) \approx p_{0,r}(k)p_{x,r}(k)$. Because for large values of k , $p_{0 \cap x, r}(k) \approx p_{0,r}(k) \approx p_{x,r}(k) \approx 1$, we say that the visitation events are effectively independent when the relative

FIG. 1. Distribution $1 - p_{x,r}(k)$ of the probability not to have visited site x among the k γ -incursions ($\gamma = 1$) inside a ball of radius $r = 100/e$, $r = 1000/e$ and $r = 10000/e$ (from left to right), and for $x = 0$, $x = 10$, $x = 20$ (blue stars, orange circles, and green squares, respectively).

difference

$$C_r^k(x) = \frac{|p_{0 \cap x,r}(k) - p_{0,r}(k)p_{x,r}(k)|}{1 - p_{0 \cap x,r}(k)} \quad (1)$$

is smaller than a threshold value (0.1 for example). In Fig. 2, we show that for increasing values of r , for both $k = 1$ and $k = 10$, the fraction of sites x whose visitation is effectively independent of the visitation of site 0 increases. This is a direct consequence of the decrease of $C_r^k(x)$ with both x/r and r .

FIG. 2. Relative difference $C_r^k(x)$ of the simultaneous visitation probability of site 0 and site x for $k = 1$ (left) and $k = 10$ (right), for γ -incursions ($\gamma = 1$) inside a ball of radius $r = 10^2/e$, $r = 10^4/e$, and $r = 10^6/e$ (blue stars, orange circles, and green squares, respectively).

Finally, as shown in the SI, (a) and (b) lead to the scaling $\rho_n \propto n^{1/2d_f}$ for the marginal case. As noted by the Referee, the scaling presented in Figure 2 of the SI is not perfect. This is due to: (i) the fact that, as stated above, (a) and (b) are only asymptotically valid (when $r \rightarrow \infty$), and (ii) logarithmic corrections with n in the scaling of ρ_n , which we leave aside in this work (as stated explicitly in the manuscript, see caption of Table 1).

As a final point, we would like to note that, as stated in the SI, this scaling of ρ_n has already been obtained in the mathematical literature (see reference S4 of the SI) in the particular case of a simple random walk in dimension $d = 2$ (which is another realization of the marginal case). This provides an additional confirmation of our result for ρ_n in the general marginal case.

By numerically characterizing more direct quantities of the visitation process during γ -incursions, we hope that we have successfully responded to the concerns of Referee 1.